# Characterization of Three New Outer Membrane Adhesion Proteins in *Fusobacterium necrophorum*

**DOI:** 10.3390/microorganisms11122968

**Published:** 2023-12-12

**Authors:** Prabha K. Bista, Deepti Pillai, Sanjeev K. Narayanan

**Affiliations:** 1Department of Comparative Pathobiology, Purdue University, West Lafayette, IN 47907, USA; pbista@purdue.edu (P.K.B.); pillai6@purdue.edu (D.P.); 2Indiana Animal Disease and Diagnostic Laboratory, Purdue University, West Lafayette, IN 47907, USA

**Keywords:** *Fusobacterium necrophorum*, liver abscess, outer membrane proteins (OMPs), OmpH, OmpA, cell surface protein

## Abstract

*Fusobacterium necrophorum,* an anaerobic Gram-negative pathogen, causes necrotic cattle infections, impacting livestock health and the US feedlot industry. Antibiotic administration is the mainstay for treating *F. necrophorum* infections, although resistance hampers their effectiveness. Vaccination, especially targeting outer membrane proteins (OMPs) due to their antigenic properties and host specificity, offers an alternative to antibiotics. This study identified high-binding-affinity adhesion proteins from *F. necrophorum* using binding and pull-down assays with bovine adrenal gland endothelial cells (EJG). Four OMP candidates (17.5 kDa/OmpH, 22.7 kDa/OmpA, 66.3 kDa/cell surface protein (CSP), and a previously characterized 43 kDa OMP) were expressed as recombinant proteins and purified. Rabbit polyclonal antibodies to recombinant OMPs were generated, and their ability to inhibit bacterial binding in vitro was assessed. The results show that treatment with individual polyclonal antibodies against 43 kDa significantly inhibited bacterial adhesion, while other antibodies were less potent. However, combinations of two or more antibodies showed a more prominent inhibitory effect on host-cell adhesion. Thus, our findings suggest that the identified OMPs are involved in fusobacterial attachment to host cells and may have the potential to be leveraged in combination for vaccine development. Future in vivo studies are needed to validate their roles and test the feasibility of an OMP-based subunit vaccine against fusobacterial infections.

## 1. Introduction

Liver abscess is a significant problem in the feedlot industry, leading to poor animal health and substantial economic losses for cattle owners. This condition is caused by the Gram-negative bacterium *Fusobacterium necrophorum* [1], an opportunistic anaerobic pathogen that produces various necrotic infections, including foot rot and calf diphtheria [2,3,4]. The incidence of liver abscesses in United States (US) feedlots is highly variable, ranging from 10% to 20% annually [5]. Critically, liver abscesses are responsible for approximately 17–32% of liver condemnations in the US, making this condition a key contributor to the overall liver condemnation rate of 45%, as reported by the 2022 National Beef Quality Audit, which has remained constant since the 2016 audit [6,7]. In severe cases, rupture of the abscessed liver during trimming requires the entire carcass to be condemned, resulting in further monetary losses and imposing a substantial economic burden on the feedlot industry.

Antibiotic prophylaxis has effectively reduced the severity of liver abscesses [8], with the US Feed Additive Compendium listing five antimicrobials currently approved for use to prevent this disease in cattle [9]. Among these drugs, tylosin is the most effective antibiotic, but even it does not entirely prevent liver abscesses [8,10]. In addition, there has been a surge in antibiotic resistance in recent years, a process that we now know is inevitable [11]. These rising resistance rates, combined with increased public health concern about the use of antibiotics for growth promotion in food animals and the growing consumer demand for “No added antibiotics”, have driven the search for viable alternatives to antibiotics. Given that liver abscess is diagnosed only at the late stages of the disease, leading to poor prognoses for infected animals [10,12], the best option is likely to be the development of a potent vaccine that effectively prevents *F. necrophorum* infection in cattle.

Outer membrane proteins (OMPs) are integral membrane proteins with a β-barrel fold, ranging in size from 8 to 36 β-strands arranged in an antiparallel pattern [13]. These molecules are anchored to lipids within the outermost bacterial membrane—a structure that has multiple essential functions, including the maintenance of cell structure, nutrient uptake, cell signaling, cell adhesion, and waste removal [14,15]. In pathogenic bacteria, OMPs play vital roles in pathogenesis, serum resistance, and disease development via their ability to mediate nutrient scavenging, promote the evasion of host defense mechanisms, and act as a channel for siderophore receptors, proteases, and lipase enzymes [16]. In fusobacterial infections, OMP-mediated adhesion and invasion is an acknowledged pathogenesis strategy [17]. OMPs are considered to have an essential immune evasion role in many bacteria, including *F. nucleatum* [18,19]. These proteins are also ideal vaccine candidates due to their exposed epitopes on the bacterial surface, which are highly conserved and readily recognized by the host immune system [20]. A unique facet of this research involves modern bioinformatics and in silico investigations, which place greater emphasis on exploring OMPs and secretory elements. This emphasis highlights these components as promising candidates in identifying vaccine targets [21,22]. OMP-mediated adherence is often the first step in infection, and thus, inhibiting this essential interaction represents a possible infection-prevention strategy [23,24]. In the absence of an effective vaccine, the well-being of feedlot cattle remains in jeopardy, representing a substantial risk to the feedlot industry. This underscores the need for the development of a robust vaccine. In this study, we aimed to uncover potential vaccine targets by identifying *F. necrophorum* OMPs with a high binding affinity to host cells using in vitro binding assays and pull-down approaches. We then generated polyclonal antibodies to recombinant forms of four of these proteins and assessed their ability to inhibit bacterial adhesion to bovine adrenal gland endothelial cells in vitro, alone and in combination. We show that antibodies to the identified proteins can effectively inhibit host-cell binding when used in combination, and thus, these OMPs may hold potential for *F. necrophorum* vaccine development.

## 2. Materials and Methods

### 2.1. Bacterial Cultures and Conditions

The bacterial strain *F. necrophorum* subspecies *necrophorum* 8L1 was isolated from a bovine liver abscess at a farm in Concordia, KS, USA [25]. Bacterial isolates were grown at 37 °C in an anaerobic chamber on sheep blood agar plates (Thermo Fisher Scientific, Waltham, MA, USA), and pure isolated colonies were inoculated in pre-reduced anaerobically sterilized brain heart infusion (PRAS-BHI) until the cultures reached an optical density at 600 nm (OD_600_) value of 0.4–0.6. Bacterial cultures in the logarithmic phase were confirmed for their purity by matrix-assisted laser desorption/ionization (MALDI) mass spectrometry (MS). Recombinant OMP expression plasmids were transformed into the *Escherichia coli* strains BL21(DE3) and BL21 Star (DE3) pLysS (Novagen, MilliporeSigma, Burlington, MA, USA and Invitrogen, Thermo Fisher Scientific, Waltham, MA, USA). The resulting bacteria were grown aerobically at 37 °C on Luria–Bertani (LB) agar and in LB broth with 100 µg/mL ampicillin, plus an additional 34 µg/mL ciprofloxacin for BL21 Star (DE3) pLysS cells.

### 2.2. Bovine Endothelial Cell Line

Bovine adrenal gland endothelial cells (EJG cells; CRL-8659, The American Type Culture Collection, Manassas, VA, USA) for bacterial infection, binding, and pull-down assays were maintained in Eagle’s Minimum Essential Medium (EMEM) containing 1.5 g/L sodium bicarbonate, non-essential amino acids, L-glutamine, and sodium pyruvate (Mediatech Inc., Corning, Manassas, VA, USA), supplemented with 10% fetal bovine serum (FBS; #35016CV, Mediatech Inc., Corning), 1% Penicillin–Streptomycin Solution 50× (5000 IU/mL penicillin and 5000 µg/mL streptomycin; #30001CI, Mediatech Inc., Corning), and 0.1% ciprofloxacin HCl (10 mg/mL; BioPlus Chemicals, Kuala Lumpur, Malaysia). The cells were incubated at 37 °C in a CO_2_ incubator and grown to 80% confluency. Confluent cells were split using 0.25% trypsin (#25053CI, Corning) and passaged every 5 to 7 days.

### 2.3. OMP Extraction

The OMPs of *F. necrophorum* subspecies *necrophorum* 8L1 were extracted as described by Osborn and Munson with some modifications [26]. In brief, a bacterial culture was grown to the logarithmic phase in 1 L of PRAS-BHI in an anaerobic chamber at 37 °C. Cells were pelleted by centrifugation (1200× *g* for 15 min at 4 °C) and resuspended in 20 mL of cold 0.75 M sucrose–10 mM Tris buffer at pH 7.8, with lysozyme added at a final concentration of 2 mg/mL of cell suspension. Cells were incubated on ice for 20 min, followed by the slow addition of two volumes of cold 1.5 mM EDTA at a constant rate for 8–10 min with continual swirling in ice. Cells were lysed by ultrasonication in an ice water bath (3 mm microtip at the 20 W output pulse setting), and the lysate was centrifuged at 6500× *g* for 15 min at 4 °C. The supernatant was then collected and subjected to ultracentrifugation at 387,000× *g* and 4 °C for 2 h in a Beckman Coulter (Brea, CA, USA) centrifuge and SW40Ti rotor. The supernatant was discarded, and the pellet was resuspended in 2 mL of 20 mg/mL Triton X-100 and 10 mL of cold STE (0.25 M sucrose, 3.3 mM Tris, 1 mM EDTA, pH 7.8) buffer. This solution was incubated at room temperature for 45 min, after which the volume was adjusted with STE buffer, and it was ultracentrifuged at 125,000× *g* and 4 °C for 2 h. The pellet was then collected in STE buffer and stored at –80 °C.

### 2.4. Binding Assays

EJG cells were grown to confluency in 6-well plates (CRL-8659, American type culture collection, Manassas, VA, USA) and incubated overnight at 37 °C with approximately 400 μg of extracted OMPs from *F. necrophorum* subsp. *necrophorum* 8L1. The cells were then washed by vigorous shaking in sterile phosphate-buffered saline (PBS) four to five times to remove unbound OMPs, followed by two washes in buffers of increasing strength: (1) PBS containing 0.1% nonyl phenoxypolyethoxylethanol-40 (NP-40) and (2) modified radioimmunoprecipitation assay (RIPA) buffer. Finally, the OMP-bound EJG cells were scraped from the plates and resuspended in SDS-PAGE buffer. The samples were then heated at 95 °C for 10 min, and proteins were separated by SDS-PAGE using a 4% stacking gel and 10% separation gel. OMP proteins were identified by Western blot using a polyclonal antibody raised against *F. necrophorum* subsp. *necrophorum* OMPs as the primary antibody [17] (Cocalico Biologicals, Stevens, PA, USA) and an alkaline phosphatase-conjugated anti-rabbit secondary antibody (12-448, EMD Millipore Corp, Billerica, MA, USA). The corresponding bands detected by Western blot were subjected to protein sequencing via liquid chromatography–tandem MS (LC-MS/MS) (Poochon Scientific, Frederick, MD, USA).

### 2.5. Pull-Down Assays

Cell surface proteins from EJG cells were isolated using the Pierce Cell Surface Protein Isolation Kit (89881, Thermo Fisher Scientific) according to the manufacturer’s instructions. In brief, EJG cells were subjected to biotinylation, followed by cell lysis, the isolation of labeled proteins, and protein elution. Labeled cell surface proteins from EJG cells were incubated with the OMP extract of *F. necrophorum* subsp. *necrophorum* 8L1, and pull-down assays were performed using the Pierce pull-down biotinylated protein: protein interaction kit (21115, Thermo Fisher Scientific), according to the manufacturer’s instructions. In these assays, the biotinylated EJG cell surface proteins were used as the bait, and the OMP extract from *F. necrophorum* was used as the prey. The interacting OMP eluant was then analyzed by 4–12% SDS-PAGE, followed by Western blot analysis with the anti-OMP antibody. The corresponding bands detected by Western blot were subjected to protein sequencing via LC-MS/MS (Poochon Scientific).

### 2.6. Bioinformatics

The online web server packages pSORTb v.3.0.3 (https://psort.org/psortb/, accessed on 7 March 2021) and Cell-PLoc 2.0 (http://www.csbio.sjtu.edu.cn/bioinf/Cell-PLoc-2/, accessed on 10 July 2023) were used to predict protein subcellular localization, and the SignalP 4.1 server was used to identify protein cleavage sites. The signal peptide sequences predicted by the SignalP server were removed from the complete protein sequences prior to cloning and expression in *E. coli*. The results of PCR product and recombinant protein sequencing were confirmed by a homologous sequence search using the Basic Local Alignment Search Tool (BLAST)n and BLASTp servers from the National Center for Biotechnology.

### 2.7. Cloning, Expression, and Purification of Recombinant OMPs

Four high-binding-affinity OMPs identified from binding and pull-down assays were selected for further analysis. Molecular cloning was performed as described previously [27]. *BamH*I and *Pst*I restriction sites and leader sequences were introduced with each forward and reverse primer, respectively, as shown in Table 1. OMP genes were PCR-amplified with Ex *Taq* DNA-polymerase (TaKaRa Bio Inc., Kusatsu, Shiga, Japan), using an annealing temperature of 50 °C for the 17 kDa and 22 kDa proteins, 54 °C for the 43 kDa OMP, and 52 °C for the cell surface protein (CSP) gene. PCR products were confirmed by Sanger sequencing (Purdue Genomics Core, West Lafayette, IN, USA and GENEWIZ, South Plainfield, NJ, USA). The obtained PCR products were then digested with restriction enzymes (RE) *BamH*I and *Pst*I (New England Biolabs, Ipswich, MA, USA), ligated using T4 ligase (M0202S, New England Biolabs), and cloned into the pET45b(+) vector (#71327, Novagen, MilliporeSigma). The 17 kDa and 22 kDa recombinant plasmids were transformed into BL21 (DE3) competent cells (Novagen, MilliporeSigma), and the 43 kDa and CSP recombinant plasmids were transformed into BL21 Star (DE3) pLysS competent cells (#C602003, Thermo Fisher Scientific) using the heat shock method.

*E. coli* clones were confirmed by Sanger sequencing and RE digestion of the recombinant plasmids. Recombinant protein expression and purification were performed as described by Novinrooz et al., with some modifications [28]. In brief, an isolated colony of bacteria containing each recombinant plasmid was grown in LB containing the appropriate antibiotics until the culture reached an OD_600_ of 0.6. Cultures were induced with 0.4 mM Isopropyl β-D-1-thiogalactopyranoside (IPTG) for 4 h, with uninduced bacterial culture used as a control. Cells were then pelleted, resuspended and lysed in SDS-PAGE buffer, and run on a gel. Western blot analysis with an anti-His tag antibody was performed to identify recombinant proteins. Protein bands were excised and sent for LC-MS/MS sequencing (Poochon Scientific). After identification, each recombinant protein was purified under denaturing conditions using the QIAGEN Ni-NTA Fast Start Protein Purification Kit (#30600, QIAGEN, Hilden, Germany) as per the manufacturer’s guidelines. Approximately 2 mg of each purified protein obtained through dialysis was sent to Lampire Biologicals Lab (Pipersville, PA, USA) for polyclonal antibody production. In the case of the CSP, further purification was performed by Size-Exclusion Chromatography by Lampire Biologicals before immunization.

### 2.8. Enzyme-Linked Immunosorbent Assays (ELISAs)

Anti-OMP immunoglobulin G (IgG) titers were quantified using an indirect ELISA (Lampire Biologicals). In brief, 96-well ELISA plates were coated with 1 µg/well of recombinant OMP and blocked with bovine serum albumin, excluding column 12. Serum samples were added (1:100 dilution) to all wells except columns 11 (reagent blank) and 12 (plate blank) in duplicate rows, and plates were incubated at 37 °C for 1 h. The plates were washed with PBS containing 0.05% Tween 20, and a secondary antibody conjugated to horseradish peroxidase (HRP) was then added and incubated for 30 min. Plates were washed again, and ABTS (2,2′-Azinobis [3-ethyl benzothiazoline-6-sulfonic acid]-diammonium salt; MilliporeSigma) was added as the substrate for visualization; after an appropriate time, the reaction was stopped. Signals were read at a wavelength of 405 nm and interpreted with SoftMax Software; version 7.1. Antibody titers were determined as the inverse of the serum dilution that reached 50% of the maximum optical density (OD) obtained, based on the generated OD values and curves.

### 2.9. Adhesion Inhibition Assays

EJG cells were seeded at 1 × 10^5^ cells/mL in 6-well plates (#229506, Celltreat Scientific products, Pepperell, MA, USA) containing 3 mL of EMEM with 10% FBS per well and incubated for 48 h. Cells were fixed using a modified Karnovsky fixative (#15720, Electron Microscopy Sciences, Hatfield, PA, USA) and washed three times with PBS to remove the fixative before the assay. *F. necrophorum* subsp. *necrophorum* 8L1 was grown to an OD_600_ of approximately 0.6. Bacterial cells were then washed and pre-treated with anti-OMP antibodies (alone or in combination) at a 1:100 dilution of polyclonal antisera for 1 h; control bacteria were untreated or treated with pre-bleed sera. After incubation, treated bacterial cells were washed to remove unbound antibodies and then added to wells containing fixed EJG cells at a multiplicity of infection (MOI) of 100:1. After a 1 h incubation, EJG cells were washed four times with sterile PBS to remove any unbound (free) bacterial cells. EJG cells with attached bacteria were then detached using 500 μL of trypsin, followed by neutralization with 1.5 mL of anaerobic EMEM plus 10% FBS. The cells were scraped from the plate, serially diluted in LB, and plated on blood agar plates. Plates were incubated under anaerobic conditions at 37 °C, and colonies were counted after 48 h.

For visual analysis, the EJG cells were seeded in 6-well plates containing coverslips (1 coverslip/well) and grown as described above. After incubating cells with or without antibodies, pre-treated bacterial cells were washed three times with PBS and then fixed with methanol for 1 min. Cells were stained with Wright–Giemsa stain for 1 min, washed with PBS (pH 6.8) for 2 min, and washed again with water for an additional 2 min. The stained cells were then dried, observed under a microscope at a magnification of 200×, and photographed.

### 2.10. Statistical Analysis

All data were obtained from at least three independent experiments. Data were analyzed using GraphPad Prism v.8.0 for Windows (GraphPad Software), and statistical significance was determined using an unpaired two-tailed Student’s *t*-test, with *p* < 0.05 considered statistically significant.

## 3. Results

### 3.1. Identification and Characterization of OMPs from F. necrophorum subsp. necrophorum with the Ability to Bind a Host Endothelial Cell Line

Total OMPs extracted from *F. necrophorum* subsp. *necrophorum* 8L1 were used in binding assays and pull-down assays with EJG bovine endothelial cells to identify bacterial surface proteins with the ability to bind host cells. In OMP/EJG binding assays, we detected four OMPs with the ability to tightly bind EJG cells, having estimated weights of 17 kDa, 24 kDa, 40 kDa, and 74 kDa (Figure 1). Of these, the 17 kDa, 24 kDa, and 40 kDa proteins were chosen for further analysis. Pull-down assays identified an additional EJG-binding protein of 66 kDa (Figure 1). Based on LC-MS/MS analysis, the 17 kDa and 22 kDa proteins showed similarity to the OmpH (17.5 kDa) and OmpA (22.7 kDa) family proteins, respectively. In addition, the 40 kDa protein was identified as a previously characterized 43 kDa OMP [27], and the 66 kDa protein was identified as a CSP (66.3 kDa). Subcellular localization prediction using the amino acid sequences identified from LC-MS/MS protein sequencing identified each protein as an OMP (Table 2).

### 3.2. Expression and Purification of Recombinant OMPs from F. necrophorum

To generate recombinant OMPs for further analysis, PCR was used to amplify the coding regions of the OmpH, OmpA, 43 kDa, and CSP genes minus their respective signal peptides, producing DNA fragments of 654 bp, 596 bp, 1050 bp, and 1845 bp, respectively. These fragments were cloned individually into the pET45b(+) expression vector, and recombinant proteins were expressed in *E. coli* via IPTG induction. OmpH and OmpA were expressed in BL21(DE3) cells, and the 43 kDa OMP and CSP were expressed in BL21 Star (DE3) pLysS cells. The recombinant His-tagged proteins were purified using nickel-NTA chromatography. Protein purity was assessed by SDS-PAGE with Coomassie staining (Figure 2A–D(i)), and proteins were confirmed by Western blot analysis (Figure 2A–D(ii)). The complete SDS-PAGE and Western blot images of samples obtained at various purification steps are provided (Appendix A). Purified rOmpH, rOmpA, rCSP, and r43 kDa OMP produced bands with the expected molecular weights of 17.5 kDa, 22.7 kDa, 42.9 kDa, and 66.3 kDa, respectively, in SDS-PAGE and Western blot analysis.

### 3.3. Humoral Responses to Purified Recombinant Proteins in Immunized Rabbits and Immunoblotting

Rabbits were immunized with recombinant OMPs conjugated with a keyhole limpet hemocyanin (KLH) antigen carrier on days 0, 21, and 42; pre-bleed sera were harvested on day 0 as the negative-control bleed. Following the first and second immunizations, test bleeds were harvested on day 30, and antibody titers were determined by ELISA. The results show that titers for each immunization were in the range of (1:10,000–100,000) (data provided in Appendix A). Test antisera were then harvested after the third immunization on day 50 and again in the final production bleed. After each administration, we found that IgG levels were increased significantly relative to day 0 (Figure 3A–D).

Rabbit sera collected on days 30 and 50 were used as primary antibodies in Western blot analysis to confirm the production of antibodies specific for each recombinant protein. The results of these assays confirm the detection of protein bands with molecular weights of 17.5 kDa, 22.7 kDa, 43 kDa, and 66.3 kDa, corresponding to OmpH, OmpA, a 43 kDa OMP, and a CSP, respectively (Figure 3E–H). All protein sizes agree with the predicted molecular weights of the recombinant proteins.

### 3.4. In Vitro Adhesion Inhibition Assay

Adhesion inhibition assays were conducted to evaluate the impact of antibody treatment on *F. necrophorum* binding to host cells. When targeting individual antibodies against OmpH or OmpA, there was no significant reduction in host-cell binding, although a trend toward reduced binding was observed. Anti-CSP antibodies displayed inconsistent effects but ultimately led to a significant reduction in bacterial binding. Notably, pre-treatment with antibodies against r43 kDa OMP significantly inhibited *F. necrophorum* binding (*p* < 0.05) (Figure 4A).

When combining anti-rOmpH and anti-rOmpA antibodies or anti-rOmpH, anti-rOmpA, and anti-rCSP antibodies, significant bacterial binding inhibition was observed, as depicted in Figure 4B,C.

Furthermore, a cocktail of antibodies directed against rOmpH, rOmpA, rCSP, and r43 kDa OMP led to substantial inhibition (Figure 4D).

The impact of antibody pre-treatment on bacterial binding was visualized through Giemsa staining. The results confirmed a reduction in the attachment of antibody-pre-treated bacteria to host cells (Figure 4E).

In summary, the use of antibody cocktails targeting bacterial OMPs effectively inhibits *F. necrophorum* binding to host endothelial cells.

## 4. Discussion

*F*. *necrophorum* is the most common cause of liver abscesses in cattle. Other than prophylactic antibiotics, vaccination is considered the most cost-effective and efficient approach for preventing this infectious disease and reducing the severity of infection. Several vaccination attempts against *F*. *necrophorum* using various bacterial virulence factors have been reported; however, only a few candidates progressed to in vivo studies, and these were found to be ineffective in generating protective immunization. In one case, a vaccine was developed targeting leukotoxin, a major virulence factor that allows *F. necrophorum* to infect and produce abscesses in the liver (with the *Trueperella pyogenes* bacterin; Centurion™ Merck Animal Health) [29,30,31]. This vaccine was shown to reduce the severity and incidence of abscesses by up to 40%; however, it is no longer commercially available. Thus, there is a continued need for the development of an efficacious vaccine to protect against *F*. *necrophorum* infection in cattle.

To support such vaccine development efforts, this study aimed to identify *F*. *necrophorum* OMPs—bacterial surface proteins that may serve as vaccine targets. In particular, OMPs with the ability to inhibit bacterial binding, as determined via antibody-blocking or serum-neutralization assays, could have potential for use in a subunit vaccine to prevent fusobacterial infection. Studies employing reverse vaccinology, in silico analysis, and bioinformatics to predict antigenic vaccine candidates have recognized outer membrane proteins as significant virulence factors. These proteins have been investigated as vaccine candidates in various bacterial species, such as *Aeromonas hydrophila* [22], *Salmonella typhi* [32], and *Klebsiella pneumoniae* [33], due to their capacity to elicit an effective immune response. Here, using host-cell-binding and pull-down assays, we identified four putative OMPs from *F. necrophorum* subspecies *necrophorum* (17 kDa, 24 kDa, 43 kDa, and 74 kDa) and one CSP (66.3 kDa), which may function as important adhesins to mediate host-cell attachment, based on results from our in vitro study.

The results of LC-MS/MS sequencing revealed that the 17.5 kDa OMP is a member of the OmpH protein family. The OmpH protein has been studied as a possible vaccine candidate against many Gram-negative pathogenic bacteria, including *Pasteurella multocida*, *Coxiella burnetti*, and *Chlamydia pneumoniae* [34,35,36]. Protein homology analysis using BLASTp revealed that OmpH is highly conserved among *Fusobacterium* species, including *F. nucleatum, F. gonidiaformans, F. hwasooki,* and *F. periodonticum,* as well as in some species of *Cetobacterium* and *Bacteroidaceae.* This protein was also found to be highly conserved within *F. necrophorum* [37]. Notably, the OmpH protein has been extensively studied in *P. multocida* and was shown to confer cross-protection to multiple strains of *P. multocida* [34,38,39]. Sequence-based structural analysis and pFam analyses further suggest that OmpH-like proteins are members of the Skp family, with chaperonin activity that is essential for the correct biosynthesis of the outer membrane.

Similarly, members of the OmpA family are essential OMPs, which have been widely studied in *Acinetobacter baumanni* [40], *Mannheimia haemolytica*, *E. coli* [41], *Haemophilus* species, and others [42,43]. In general, OmpA-like proteins sense and respond to external stimuli or stressors [44]. In *E. coli*, OmpA is involved in the expression of several virulence factors, such as fimbriae, and in biofilm formation [45]. In *K. pneumoniae,* OmpA functions as a receptor for several bacteriophages, as well as for *Shigella boydii* bacteriocins, and it was further shown to act as a pathogen-associated molecular pattern in *K. pneumoniae*-infected hosts [44,46]. OmpA-like proteins are thought to be present in high copy numbers on exposed cell surfaces, making them ideal candidates for vaccine development. In *P. multocida,* for example, anti-OmpA antibodies significantly reduce bacterial adherence to extracellular matrices [47]. Recent studies on *S. Typhi* further support the protection against typhoid fever in a mouse model generated by an antibody raised against recombinant OmpA [32]. Our BLASTp analysis revealed that OmpA family proteins are conserved in *Fusobacterium* species, with >60% amino acid sequence identity between OmpA proteins from *F. gonidiaformans*, *F. nucleatum,* and other related bacterial species (e.g., *Cetobacterium* species).

The 66.3 kDa CSP identified here in *F. necrophorum* is homologous to YadA-like family proteins and is conserved in *F*. *necrophorum* and *F. nucleatum* [48]. YadA-like proteins have been studied in several bacterial pathogens. In *Haemophilus influenzae,* YadA-like protein is a putative virulence factor, although its precise function has not been defined [49], and in *Yersinia*, the YadA domain was shown to be involved in adhesion and resistance to host defense activity [50]. Likewise, the 43 kDa OMP—the prominent OMP identified in *F*. *necrophorum*—was found to have 96% homology with the FomA protein of *F. nucleatum*. Consistent with the results of our adhesion inhibition assays, a previous study by Kumar et al. reported that FomA antibodies are highly effective at preventing the attachment of bacteria to endothelial cells [27] and as a component of effective subunit vaccines [51].

In our study, individual polyclonal antibodies raised against *F. necrophorum* OMPs showed limited efficacy in preventing bacterial adhesion to the EJG cell line in vitro, with a notable but non-significant trend toward reduced adhesion in antibody-treated cells. An exception was observed with polyclonal antisera targeting the 43 kDa OMP, which significantly inhibited bacterial binding, consistent with prior studies on FomA/43 kDa OMP, as noted above [27]. Additionally, combinations of anti-OMP antibodies demonstrated a significant capacity to inhibit *F*. *necrophorum* binding to host cells. Importantly, even the combination lacking anti-43 kDa OMP antibodies displayed a substantial inhibitory effect on binding, thus excluding the possibility that the observed effect in the four-antibody cocktail was solely attributed to the presence of 43 kDa OMP antibodies alone. These findings suggest that a multi-subunit approach may represent a promising strategy for vaccine development, meriting further investigation. Although the reason for the observed synergistic inhibitory effect with multiple OMP antibodies is not known, there are several possibilities. For example, multiple adhesins may act collaboratively via differential expression at various stages during the infection process [52]. This mechanism would also explain why treatment with individual antibodies was unable to prevent bacterial attachment. Alternatively, it is possible that the concentration of antibodies in sera used for our in vitro experiments was insufficient for neutralization. In a study using OMPs as a subunit vaccine in *E. coli* O78, three outer membrane proteins in combination were used for immunization, which resulted in higher immunogenicity [53]. Future in vivo studies are therefore needed to elucidate the precise functions of these conserved OMPs and determine whether they have the potential for vaccine development, as demonstrated in other pathogenic bacteria. In conclusion, in this study, we identified and characterized four OMPs of *F. necrophorum*, which show high binding affinity to host endothelial cells. Our results further suggest that multiple adhesins may contribute to bacterial attachment, highlighting the potential utility of using multiple OMPs in a combination subunit vaccine against *F*. *necrophorum* infections in cattle. Given that OMPs are conserved and immunogenic, these vaccines may also be effective at preventing infection by various other *Fusobacterium* species. Thus, future studies are needed to evaluate the ability of these vaccine candidates to collectively induce protective immunity in vivo.

## Figures and Tables

**Figure 1 microorganisms-11-02968-f001:**
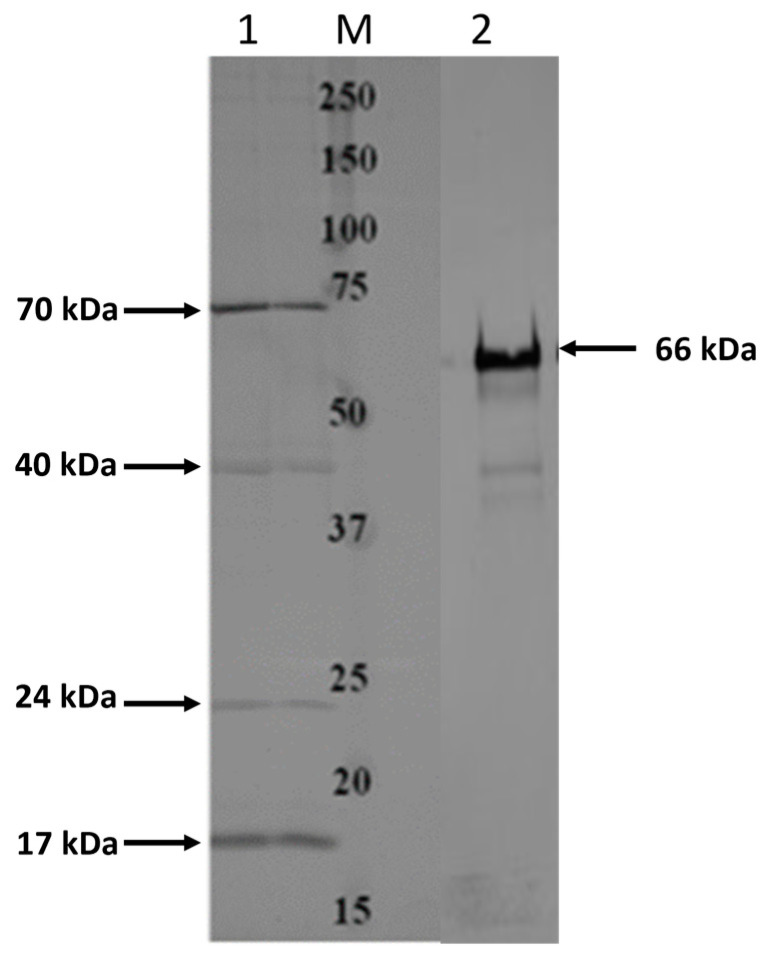
Identification of high-binding-affinity adhesins from *Fusobacterium necrophorum* subspecies *necrophorum*. Results from binding assays (adapted from [17]) and pull-down assays with *F. necrophorum* outer membrane protein (OMP) extract and EJG bovine endothelial cells. OMPs bound to host cells were identified by Western blot analysis with anti-OMP primary antibody; Lane M indicates protein marker, and Lanes 1 and 2 show the OMPs identified from binding assays and pull-down assays, respectively.

**Figure 2 microorganisms-11-02968-f002:**
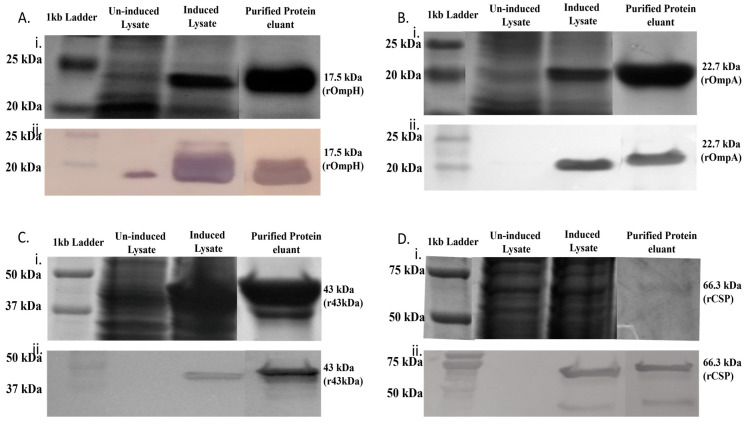
Expression and purification of recombinant high-affinity-binding proteins from *F. necrophorum* in *Escherichia coli*. (**A**–**D**) represent rOmpH, rOmpA, rCSP, and r43 kDa OMP, respectively. (**i**) depicts SDS-PAGE analysis, and (**ii**) corresponds to Western blot analysis. Western blots were incubated with anti-His primary antibody and anti-mouse IgG secondary antibody for rOmpH, rOmpA, r43kDa OMP, and rCSP, respectively. Lane 1: molecular marker; Lane 2: uninduced whole-cell lysates; Lane 3: induced whole-cell lysates; and Lane 4: purified recombinant OMPs.

**Figure 3 microorganisms-11-02968-f003:**
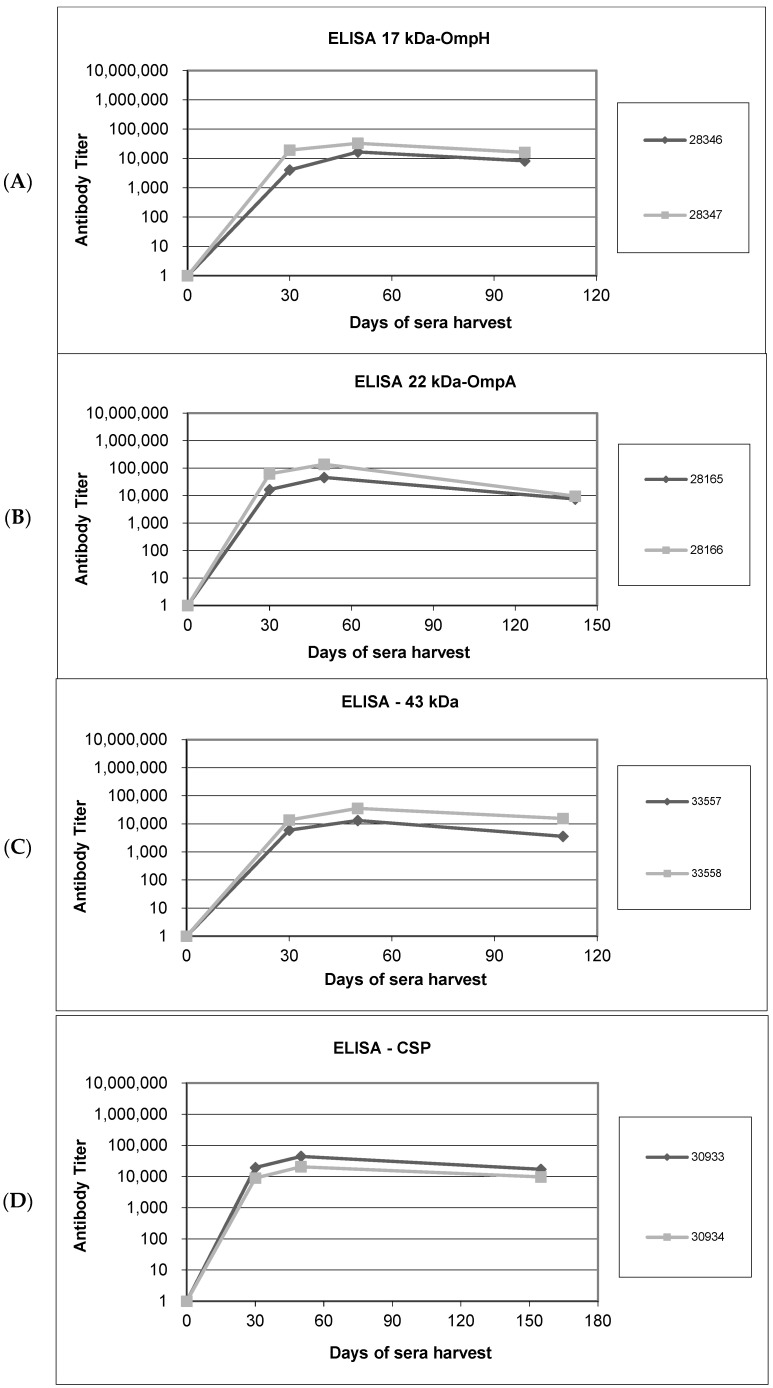
Humoral responses in rabbits immunized with recombinant antigens and Western blot analyses to confirm antibody specificity in sera harvested after immunization. (**A**–**D**). Enzyme-linked immunosorbent assays (ELISAs) to measure antibody titers in sera from rabbits immunized with recombinant OMPs: (**A**) rOmpH, (**B**) rOmpA, (**C**) r43 kDa OMP, and (**D**) rCSP. Pre-immunization antibody titers on day 0 and titers on days 30 and 50 post-immunization and on harvest day are shown. Diamond-shaped dots and square-shaped dots connecting lines in each graph represent the antibody titers from two rabbits (assigned label number) immunized with each protein. (**E**–**H**). Western blot analysis to confirm the specificity of antibodies raised against the recombinant OMP antigens: (**E**) rOmpH, (**F**) rOmpA, (**G**) r43 kDa OMP, and (**H**) rCSP. The pre-bleed lanes were probed with control sera harvested before immunization (day 0), whereas day 30 sera were used as the primary antibody for analysis of the test bleed lanes.

**Figure 4 microorganisms-11-02968-f004:**
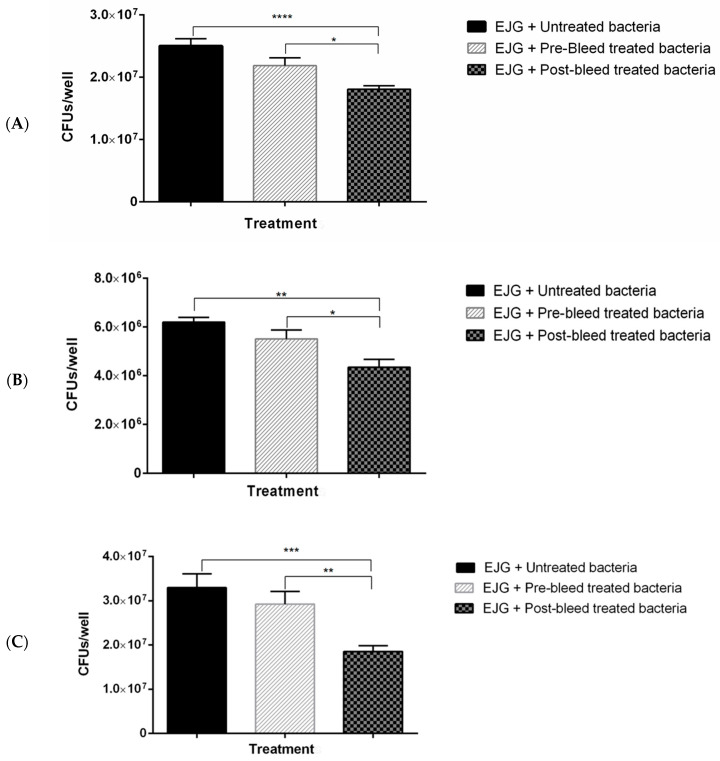
Rabbit polyclonal antibodies to recombinant OMPs inhibit adhesion of *F. necrophorum* to bovine endothelial (EJG) cells. Adhesion inhibition assays with *F. necrophorum* and EJG cells using rabbit (1:100) polyclonal antibodies raised against (**A**) r43 kDa OMP; (**B**) rOmpH and rOmpA; (**C**) rOmpH, rOmpA, and rCSP; and (**D**) rOmpH, rOmpA, rCSP, and r43 kDa OMP. Significance was determined by unpaired two-tailed Student’s *t*-test, n = 3; * *p* < 0.05, ** *p* <0.01, *** *p* < 0.001, and **** *p* < 0.0001. The bars indicate the mean ± standard error of the mean (SEM) of data pooled from triplicate experiments. (**E**) Giemsa staining of EJG cells following adhesion inhibition assays: (**i**) uninfected (negative control), (**ii**) infected with *F. necrophorum* (positive control), and (**iii**) infected with *F. necrophorum* pre-treated with a combination of four polyclonal antibodies, as in panel (**D**).

**Table 1 microorganisms-11-02968-t001:** Primers used for cloning and sequencing of recombinant plasmids.

Gene	Primer	Tm
17 kDa F	5’ TAGGATCCAGATAAAATTGCAGTGGTG 3’	56.9 °C
17 kDa R	5’ CGCTGCAGTTATTTTATTGTTTCCAT 3’	54.8 °C
22 kDa F	5’ TAGGATCCCGAAGAAGGAAATATAAAC 3’	54.3 °C
22 kDa R	5’ CGCTGCAGCTAGTTATATTTTGGAGC 3’	58.1 °C
43 kDa F	5’ TTGGGATCCTAAAGAAGTGATGCCTGCT 3’	61.5 °C
43 kDa R	5’ ACCTGCAGTTAGAAAGTAACTTTCATACC 3’	56.8 °C
CSP F	5’ GCCGGATCCAGAAGATCCGGTAATAAAAAGA 3’	61.2 °C
CSP R	5’ ACCTGCAGTTATTTGTTTATTAATTCTTC 3’	52.9 °C

Abbreviations: CSP: cell surface protein; Tm: Melting Temperature; Underlined regions indicate recognition site for restriction enzymes.

**Table 2 microorganisms-11-02968-t002:** Liquid chromatography–tandem mass spectrometry (LC-MS/MS) sequencing analysis of four *Fusobacterium necrophorum* outer membrane proteins (OMPs) identified from binding assays and pull-down assays with EJG bovine endothelial cells.

Accession	Description	# AAs	MW (kDa)
WP_035904032.1	OmpH family protein	159	17.5
WP_005961133.1	OmpA family protein	217	22.7
JQ740821.1	43 kDa outer membrane protein	377	42.9
WP_035916891.1	Cell surface protein	638	66.3

## Data Availability

All data has been included in both the manuscript and Appendix A.

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
