# Peer review of "Characterization of Three New Outer Membrane Adhesion Proteins in *Fusobacterium necrophorum"

_microorganisms, 2023, doi:10.3390/microorganisms11122968_

Round 1

Reviewer 1 Report

Comments and Suggestions for Authors

Generally, The whole writing quality of the manuscript is well, relevant, interesting, and important for the field. The article has a scientific and practical impact. I think that this manuscript could be accepted but needs minor modifications and attention before publication. 

Some minor revision suggestions are listed below: I´ll only highlight some general comments:

Introduction: Seems well and informative. however,

1- Check journal recommendations for References and P values, be consistence in this regard. 

2- Please double-check the full name of the abbreviations in their first mentions and throughout the manuscript.

3- The introduction of the manuscript is diluted and semi-adequate with respect to reviewing the latest literature relative to the study described and should be improved somewhat.

4- Please emphasize the added value/novelty of this article. 

5- Please provide more recent literature reviews and citations. 

6- The hypothesis is not clear, and the objectives in the introduction do not completely fit each other. Please clarify and recheck it again. I would add a few sentences explaining what additional information your study will bring.

- The statistical analysis explanation is not considered adequate, please explain with more details.

Results:

- The results section needs to be shortened with emphasis on important findings.

- I suggest omitting all sentences with no significant effect observed. 

- I would suggest making every effort to explain more the results and not compare them with other studies.

Discussion: is well written. However, some references are old. Can you confirm this with more recent references?

Conclusion: Seems informative and concise.

References:  There are no references reflecting 2022-2023, it should be for reference updating.

Author Response

Generally, The whole writing quality of the manuscript is well, relevant, interesting, and important for the field. The article has a scientific and practical impact. I think that this manuscript could be accepted but needs minor modifications and attention before publication. 

Some minor revision suggestions are listed below: I´ll only highlight some general comments:

Introduction: Seems well and informative. however,

  • Check journal recommendations for References and P values, be consistence in this regard. 
  • Complied to reviewer’s comments, the authors thoroughly reviewed all the references and p-values for accuracy.
  • Please double-check the full name of the abbreviations in their first mentions and throughout the manuscript.
  • Complied to reviewer’s comments, the authors conducted a comprehensive review to confirm that all abbreviations have been initially mentioned in the text and subsequently used throughout the document.
  • The introduction of the manuscript is diluted and semi-adequate with respect to reviewing the latest literature relative to the study described and should be improved somewhat.
  • Complied to reviewer’s comments, the recent supportive studies and current information has been duly acknowledged through citation, with the incorporation of additional pertinent information (highlight in the text in the following section -introduction, discussion and references).
  • Please emphasize the added value/novelty of this article. 
  • Complied to the reviewer’s comment, the novelty of the study has been addressed in Line 64-72
  • Please provide more recent literature reviews and citations.
  • Complied to reviewer’s comments, the recent literature supporting the study have been added and mentioned in Lines 43,65-66, 386-390, 417-419,451-453. (might change based on edits; highlight in the text)
  • The hypothesis is not clear, and the objectives in the introduction do not completely fit each other. Please clarify and recheck it again. I would add a few sentences explaining what additional information your study will bring.

- Complied to reviewer’s comments, the issues have been addressed in Line 63-71

- The statistical analysis explanation is not considered adequate, please explain with more details.
- Complied to reviewer’s comments, the authors included necessary statistical details in section 3.4 and in fig 4 legend.

Results:

- The results section needs to be shortened with emphasis on important findings.
- I suggest omitting all sentences with no significant effect observed. 
- I would suggest making every effort to explain more the results and not compare them with other studies.

- Complied to the reviewer’s comments, the changes have been made as suggested.

Discussion: is well written. However, some references are old. Can you confirm this with more recent references?
Complied to the reviewer’s comments, new supporting references have been added to the discussion section.

Conclusion: Seems informative and concise.

References:  There are no references reflecting 2022-2023, it should be for reference updating.
Complied to reviewer’s comment recent supporting references have been updated.

Reviewer 2 Report

Comments and Suggestions for Authors

This study has certain significance, and the experimental results are helpful to evaluate the antibacterial ability of the protein in vitro. However, the article still needs extensive revision before it can be accepted for publication.

Tm must be separated in table1.

Line 147, reference format must be uniform.

fig1A Image quality is poor and not clear. fig2A needs to be changed, the layout is confused and the picture is blurred.

In the ELISA experiment, only two individuals are used for the experiment, is there less biological duplication? The final determination time of different antibodies is not consistent, why? How was it determined? Is the host status (age, sex, etc.) consistent? Do hosts of different ages and genders influence the expression of recombinant proteins? Necessary host information is missing, including age, sex, etc.

fig4 results did not indicate the concentration and dosage of antibody. In addition, the four antibodies can be combined in a variety of ways, why choose this combination? Can we further explore the optimal combination of the four antibodies?

Author Response

This study has certain significance, and the experimental results are helpful to evaluate the antibacterial ability of the protein in vitro. However, the article still needs extensive revision before it can be accepted for publication.

Tm must be separated in table1.

  • Complied to reviewer’s comment, Tm has been placed in a separate column in Table 1

Line 147, reference format must be uniform.

  • Complied with the reviewer’s comment, the reference in the line was a mistake, and a link has been added to the sentence for the tool used for analysis.

fig1A Image quality is poor and not clear. fig2A needs to be changed, the layout is confused, and the picture is blurred.

  • Complied to reviewer’s comment ,the quality of Fig 1A and Fig 2 were enhanced through the use of image editing software, achieving a pixel density of 300 DPI.

In the ELISA experiment, only two individuals are used for the experiment, is there less biological duplication? The final determination time of different antibodies is not consistent, why? How was it determined? Is the host status (age, sex, etc.) consistent? Do hosts of different ages and genders influence the expression of recombinant proteins? Necessary host information is missing, including age, sex, etc.

- In the ELISA experiment, the main goal was to generate polyclonal antibodies against recombinant OMPs. The samples were sent to Lampire Biological Laboratories (https://www.lampire.com/services/rabbit) for their polyclonal antibody production service, where the laboratory utilized two rabbits for each OMP. It's important to note that any other metadata deemed irrelevant to the study was not considered.

fig4 results did not indicate the concentration and dosage of antibody. In addition, the four antibodies can be combined in a variety of ways, why choose this combination? Can we further explore the optimal combination of the four antibodies?

  • The concentration utilized for inhibition has been included in the legend on Line 362, and it is also detailed in the methodology section on Line 213. Given that this study involved polyclonal antibodies, specific concentration measurements, as typically conducted for monoclonal antibodies, were not performed. Therefore, a 1:100 dilution was chosen, a decision supported by previous studies such as Kumar et al., 2013. For future studies, it would be valuable to investigate the testing of various combinations of OMP subunits for antibody production.